# Potential Role of Antibodies against Aquaporin-1 in Patients with Central Nervous System Demyelination

**DOI:** 10.3390/ijms241612982

**Published:** 2023-08-19

**Authors:** Maria Pechlivanidou, Konstantina Xenou, Dimitrios Tzanetakos, Emmanuel Koutsos, Christos Stergiou, Elisabeth Andreadou, Konstantinos Voumvourakis, Sotirios Giannopoulos, Constantinos Kilidireas, Erdem Tüzün, Georgios Tsivgoulis, Socrates Tzartos, John Tzartos

**Affiliations:** 1Tzartos NeuroDiagnostics, 11523 Athens, Greece; mpechlivanidou@neurodiagnostics.gr (M.P.); kxenou@neurodiagnostics.gr (K.X.); mkoutsos@neurodiagnostics.gr (E.K.); cstergiou@neurodiagnostics.gr (C.S.); stzartos@neurodiagnostics.gr (S.T.); 2Second Department of Neurology ‘’Attikon’’ University Hospital, School of Medicine, National & Kapodistrian University of Athens, 12462 Athens, Greece; tzanetakosdim@yahoo.com (D.T.); cvoumvou@outlook.com (K.V.); sgiannop@uoi.gr (S.G.); tsivgoulisgiorg@yahoo.gr (G.T.); 3First Department of Neurology, ‘’Aiginiteion’’ University Hospital, National and Kapodistrian University of Athens, 11528 Athens, Greece; elisandread@gmail.com (E.A.); kildrcost@med.uoa.gr (C.K.); 4Second Department of Neurology, Henry Dunant Hospital Center, 11526 Athens, Greece; 5Department of Neuroscience, Aziz Sancar Institute of Experimental Medicine, Istanbul University, 34093 Istanbul, Turkey; drerdem@yahoo.com; 6Department of Neurobiology, Hellenic Pasteur Institute, 11521 Athens, Greece; 7Department of Pharmacy, University of Patras, 26504 Patras, Greece

**Keywords:** aquaporins (AQPs), AQP4, AQP1, antibodies, autoimmunity, CNS demyelination, neuromyelitis optica spectrum disorder (NMOSD), autoimmune astrocytopathy

## Abstract

Aquaporins (AQPs; AQP0–AQP12) are water channels expressed in many and diverse cell types, participating in various functions of cells, tissues, and systems, including the central nervous system (CNS). AQP dysfunction and autoimmunity to AQPs are implicated in several diseases. The best-known example of autoimmunity against AQPs concerns the antibodies to AQP4 which are involved in the pathogenesis of neuromyelitis optica spectrum disorder (NMOSD), an autoimmune astrocytopathy, causing also CNS demyelination. The present review focuses on the discovery and the potential role of antibodies against AQP1 in the CNS, and their potential involvement in the pathophysiology of NMOSD. We describe (a) the several techniques developed for the detection of the AQP1-antibodies, with emphasis on methods that specifically identify antibodies targeting the extracellular domain of AQP1, i.e., those of potential pathogenic role, and (b) the available evidence supporting the pathogenic relevance of AQP1-antibodies in the NMOSD phenotype.

## 1. Introduction

### 1.1. Aquaporin Function, Structure, and Tissue Distribution

Aquaporins (AQPs) represent an important group of transmembrane proteins, having a key role in the transport of water across biological membranes acting as a water channel [1,2]. AQPs are vital in maintaining water balance in cells and tissues demonstrating high selectivity, thus mostly allowing the transfer of water molecules, while excluding most of the other molecules and ions [1,2,3]. This selectivity is achieved through a combination of size exclusion and electrostatic interactions between the aquaporin channel and the water molecules [1,2]. 

In mammals, 13 distinct types of AQPs have been identified (AQP0 to AQP12), which are classified into three families based on their permeability status: (a) the AQPs (AQP0, AQP1, AQP2, AQP4, AQP5, AQP6, and AQP8), (b) the aquaglyceroporins (AQP3, AQP7, AQP9, and AQP10) which are additionally permeable to urea and glycerol, and (c) the ‘‘unorthodox’’ AQPs (AQP11 and AQP12) [1,2,3]. Several AQPs (i.e., AQP1, AQP3, AQP6, AQP7, AQP8, and AQP9) have been reported to be permeable to ammonia; however, the mechanism by which ammonium is transported by them has not yet been fully resolved [4].

AQPs assemble as tetramers in the cell membrane which are present in various tissues such as the kidneys, lungs, and brain [1,3,5]. In Table 1, the known types of AQPs and their main tissue distribution and function are summarized. 

### 1.2. AQPs in Disease 

Defects and/or mutations in aquaporin-related genes can lead to various diseases. For example, mutations in the AQP2 gene can cause nephrogenic diabetes insipidus (NDI) [11], in which the kidneys are unable to respond to the antidiuretic hormone vasopressin, resulting in the inability to concentrate urine, and consequently to excessive thirst and urination [12]. 

Furthermore, there is ongoing research for the development of small molecule inhibitors targeting AQPs to affect their function; an example of this category of drugs is the tolvaptan, a selective vasopressin V2 receptor antagonist that targets AQP2 in the kidneys. By blocking the interaction between vasopressin and AQP2, tolvaptan reduces water reabsorption in the collecting ducts, thereby decreasing urine concentration and increasing urine volume. It has been used for the treatment of conditions including hyponatremia and polycystic kidney disease [13]. Other AQPs may also be candidates for novel drug development, e.g., AQP4 and AQP1 which are involved in water transport across the blood–brain barrier (BBB) and play a role in brain edema [14].

### 1.3. Autoimmunity to AQPs

Antibodies against AQP channels are involved in the pathogenesis of autoimmune disorders of the central nervous system (CNS) [15], blood [16] and kidneys [17] and other tissues and organs [18]. In the CNS, the best-known example of AQP antibodies (AQP-Abs) are the antibodies against AQP4, which are associated with the majority of neuromyelitis optica spectrum disorder (NMOSD) cases, an inflammatory, antibody-mediated astrocytopathy resulting in secondary demyelination of the CNS [19,20]. 

#### AQP4 Serum Antibodies in NMOSD Phenotype

AQP4 is the most abundant aquaporin in the CNS, found in the optic nerve [21], brain [1,22], and spinal cord [23]. It is expressed primarily in perivascular astrocyte foot processes and the glial-limiting membrane [5,8,22,24], which surround blood vessels and form the BBB [22,25]. AQP4 has a vital role in cerebrospinal fluid (CSF) absorption, circulation, and homeostasis [22,25] and is the primary autoantigen in NMOSD. The initial study that reported the detection of serum antibodies in NMOSD cases was in 2004 by Lennon et al. [26], highlighting its importance as a distinct NMO-IgG serological marker, targeting AQP4 [27]. These antibodies had displayed a characteristic immunohistochemical pattern of binding to tested mouse CNS tissues [26]. The staining pattern was absent in the choroid plexus region, yet it was prominent in the micro vessels of the white and grey matter in the cerebellum, midbrain, and spinal cord. This unique staining pattern was clearly absent in control samples from patients with MS profile, underscoring the specificity of these autoantibodies for NMOSD [26]. Notably, double labelling experiments demonstrated colocalization between the NMO-associated antigen and an astrocytic marker, pinpointing the astrocytic foot processes as the region of the co-staining and suggesting that NMO-IgG could be localized at the BBB.

NMO, also known as Devic’s disease, is a relapsing-inflammatory autoimmune astrocytopathy, more commonly associated with longitudinally extensive transverse myelitis (LETM) and optic neuritis [28]. The clinical characteristics of NMOSD include optic neuritis, acute myelitis, area postrema syndrome, acute brainstem syndrome, symptomatic narcolepsy, acute diencephalic clinical syndrome, and symptomatic cerebral syndrome [15,28]. NMOSD can be misdiagnosed as MS due to overlapping clinical manifestations and radiological features [29]. The discovery of the specific serum antibody against AQP4 led to the identification of NMOSD and the use of appropriate therapies, different from the MS-specific immunotherapies [26,29,30,31]. On the other hand, absence of AQP4 antibodies in a substantial proportion of NMOSD patients has prompted the search for novel autoantibodies directed against additional astrocytic antigens.

Several reports have shown that, in addition to AQP4, human CNS astrocytes also abundantly express AQP1 on their surface [10,32]. Specifically, it has been shown that AQP1 is highly expressed in CNS regions prone to develop NMOSD lesions (spinal cord, optic nerves, and brain white matter) [29,33,34] and is involved in blood–CSF barrier functions [1,8,9,22]. Moreover, both AQP1 and AQP4 are overexpressed in the brain in some neurological diseases, such as MS, compared to ‘‘normal’’ brains [32], possibly in order to maintain water homeostasis [22]. These preliminary findings have drawn attention to the involvement of AQP1 in NMOSD and autoimmune demyelinating disorders of the brain. The present review aims to provide a concise overview of the current understanding of AQP1 autoimmunity, particularly in NMOSD, focusing on the reliable detection of antibodies that bind to AQP1 protein while it is embedded in the membrane of living cells. We sought to evaluate the frequency of AQP1-Abs in NMOSD, and their possible implication in the disease development. 

## 2. AQP1

### 2.1. The Structural Biology of AQP1 Water Channel

AQP1 is the first identified member of the aquaporin superfamily. The structure of AQP1 has been extensively studied using X-ray crystallography and electron microscopy [35]. The AQP1 monomer consists of six transmembrane alpha-helices, labeled A–F, connected by five loops, labeled Loops A–E (Figure 1). Helices A and B, as well as helices E and F, are arranged parallel to each other, forming a central channel through which water molecules diffuse [35,36]. The channel is lined with hydrophilic residues, primarily asparagine and threonine, which create a pathway that is selective for water while excluding ions and other solutes [35,36]. The loops connecting the helices extend into the cytoplasm and the extracellular space and contain sites for post-translational modification and protein–protein interactions [35,36]. The N- and C-termini of the protein are also located in the cytoplasm. AQP1 functions as a homo-tetramer, with four monomers assembling to form a functional channel [35,36]. The tetramer is square-shaped and is stabilized by interactions between the extracellular loops [35,36]. 

### 2.2. The Role of AQP1 in the Human Body 

AQP1 is expressed in numerous tissues of the body. In the kidney, AQP1 is abundant in the proximal tubules and is particularly important for water reabsorption from the urine back into the bloodstream, regulating fluid balance, retaining necessary fluids, and preventing excessive water loss. [1]. In the lungs, AQP1 aids in the movement of water vapor during respiration, maintaining optimal lung function [38,39]. In red blood cells (RBCs), AQP1 contributes to the regulation of the cell volume and the prevention of hemolysis [40]. AQP1 is also expressed widely in vascular endothelia and cardiomyocytes, where it facilitates transepithelial water transport [41]. Furthermore, AQP1 has been involved in cell migration and angiogenesis [42], as it has been shown to facilitate kidney endothelial cell migration [43], which is crucial for various physiological processes such as wound healing [44] and renal injury repair [45,46]. In the CNS, AQP1 is primarily located in the epithelial cells of the choroid plexus [1,3,22,24,47], which are a key component of the blood–CSF-barrier that controls CSF production and the exchange of nutrients and waste products between the blood and the brain [1,8,9,22,25,48]. However, in the human brain, AQP1 is also localized in the subcortical white matter (WM) [10], whereas it is much less prominent in the grey matter (GM) area of the cortex [49]. At the cellular level, human astrocytes express both AQP4 and AQP1 mRNA and proteins [10,32]. AQP1 is essential for the maintenance of water homeostasis in the CNS [1,3,22], as it facilitates water diffusion across cell membranes, which is crucial for preventing the accumulation of fluid excess that can lead to edema [3,25,48]. In the spinal cord, AQP1 in addition to astrocytes [10,32], is expressed in the ependymal cells lining the central canal, but more robustly in the sensory fibers of the superficial laminae of the dorsal horn [50].

### 2.3. Autoimmunity to AQP1 in Non-CNS Diseases 

AQP1 antibodies have been detected in autoimmune hemolytic anemia (AΙHA) and in Sögren’s syndrome (SS).

AQP1 contains the Colton blood group antigen expressed on erythrocytes [51]. Antibodies against Colton antigens are very rare and can cause transfusion reactions or hemolytic disease [52]. Interestingly, in Section 3.2 we describe a case with high titer of AQP1 antibodies and the development of both autoimmune hemolytic anemia (AIHA) and a clinical relapse in a patient with NMOSD phenotype; recovery of both diseases occurred with a simultaneous dramatic drop of AQP1-Abs.

SS is a chronic autoimmune disease that primarily targets the lacrimal and salivary glands causing dryness in the mouth and eyes. AQPs seem to play a role in salivary gland secretions/function. Alam et al. [53] reported the detection of serum antibodies against AQP5 and AQP1 in patients with SS, and since AQP1 is closely related to AQP5 in the phylogenic tree of human AQPs, the authors also sought to investigate the possible presence of anti-AQP1 autoantibodies in the sera of SS patients [54]. The authors, by screening 112 SS sera, detected anti-AQP1 antibodies in 27.7% of the SS sera but in none of the control sera [54]. Nearly half of the anti-AQP1 positive were also tested positive for anti-AQP5 IgG [54]. The authors suggest that AQP1-Abs in SS patient sera may be cross-reactive and produced during antibody response to AQP5 [54].

Furthermore, Tzartos et al. screened 34 sera from SS patients for binding to the extracellular domains of various AQPs [18]. Thirteen (38.2%) SS patients had antibodies to extracellular domains of AQPs, one of which to AQP1, but none to AQP5 [18]. Each patient had antibodies to only one extracellular domain. Interestingly, patients with antibodies to AQPs had significantly more severe xerophthalmia, compared with AQP-negative patients, suggesting a potential pathogenic role of these antibodies [18]. 

## 3. AQP1 Autoimmunity in NMOSD Phenotype 

Human CNS astrocytes express both AQP4 and AQP1 on their surface [32]. It is well known that in patients with “NMOSD phenotype” (i.e., patients with at least one of the core criteria for NMOSD), serum antibodies against AQP4 are detected frequently as we described above, while in some AQP4-Ab^neg^ cases antibodies against myelin oligodendrocyte glycoprotein (MOG) can be detected as well. It must be pointed out that the diagnosis of AQP4-Ab^neg^/MOG-Ab^neg^ (seronegative) NMOSD phenotype remains extremely challenging.

In 2013, Tzartos et al. [55] used various methods to detect serum antibodies against AQP1 in patients with NMOSD phenotype including the core clinical characteristics of LETM and/or optic neuritis, according to the 2015 NMOSD criteria [15]. Below we describe these methodologies of Tzartos et al. [55] and also of another four research groups [56,57,58,59] for the detection of AQP1-Abs in patients with NMOSD phenotype. Additionally, we discuss the results, advantages, and disadvantages of each method. Subsequently, we present the available evidence suggesting these antibodies might play a role in the pathogenesis of NMOSD.

### 3.1. Assays for the Detection of Antibodies against AQP1

Although a gold-standard method for detecting AQP1-Abs is not yet available, several different methodologies have been proven successful in detecting AQP1-Abs, which confirm one another. These include radioimmunoprecipitation assay (RIPA) with I^125^-labelled intact AQP1, Western-blot with SDS-denatured AQP1 polypeptide, ELISA with intact AQP1, ELISA with synthetic peptides corresponding to the extracellular and cytoplasmic loops of AQP1 channel, and importantly, an indirect live cell-based assay (indirect CBA) with similar reliability with the classical CBA, which is used for the detection of antibodies to several membrane antigens. Yet, probably due to the reasons explained below, a reliable classical (live) CBA has not been established until now.

#### 3.1.1. Assays for Antibody Binding to Cell-Free AQP1 

Radioimmunoprecipitation assay (RIPA)

Initially, a sensitive RIPA for AQP1-Abs was developed, using biotinylated human AQP1 indirectly labeled with I^125^-streptavidin. This test identified 58/348 (17%) patients with suspected NMOSD phenotype as positive for AQP1-Abs (AQP1-Ab^pos^ sera) (Figure 2). The antibodies of the positive sera did not bind to plain 125I-streptavidin, confirming their binding to the AQP1 itself. Interestingly, the frequency of AQP1-Ab^pos^ patients was somewhat higher than that of the AQP4-Ab^pos^ patients (12%) in the studied group. No AQP1-Abs were detected in the 242 control sera [55] and 4% (14 sera) were double-positive, i.e., with antibodies to both AQPs; it was shown that these sera had distinct antibodies to each AQP, rather than cross-reactive antibodies. This profile of the double-positivity (AQP1-Ab^pos^/AQP4-Ab^pos^) was also detected in some patients with definite NMOSD phenotype [55]. However, subsequent experiments suggested that this double positivity was due to the fact that the RIPA, with intact and soluble AQP, detected antibodies to both the extracellular (potentially pathogenic) and cytoplasmic sides (unlikely to be pathogenic) of both AQPs.

b.Western blotting with SDS-denatured AQP1 polypeptides

The RIPA results were then validated through Western blotting. All 4 AQP1 Ab-containing sera which were tested, were bound to electrophoresed SDS-denatured yeast-expressed AQP1 (Figure 3). This approach provided an alternative assay for AQP1-Abs, visualized antibody binding, and interestingly showed that AQP1 epitopes are not conformation-dependent, since it was revealed that serum autoantibodies also bind to denatured AQP1 [55].

c.ELISA with intact AQP1

Subsequently, an ELISA with immobilized purified yeast-expressed AQP1 was developed to confirm the RIPA results. Indeed, there was a very good correlation between the results of RIPA and ELISA: Therefore, because of its simplicity and similarly high specificity and sensitivity compared to RIPA, we subsequently adopted AQP1-ELISA for routine testing [55]. 

The use of AQP1-ELISA for the detection of AQP1-Abs in patient sera was also attempted by at least another two groups. Sánchez Gomar et al. [30], in addition to CBA (see below), also attempted ELISA with small groups of patients with various disease phenotypes. Although a few sera showed positive values of AQP1-Abs, the overall comparative analysis of AQP1-Ab titer among all groups revealed no statistical differences. However, the numbers of sera in the candidate groups were too small and inadequate, whereas the few detected AQP1-Ab^pos^ sera were not provided in their paper (apart from the average values for each group).

Jasiak-Zatońska et al. [59] using a commercially available ELISA kit (Human AQP1 ELISA Kit, MBS262447, MyBiosource Inc., Eersel, Netherlands) claimed that patients with MS diagnosis had higher AQP1-Ab levels than those with NMOSD phenotype. However, to our understanding, the specific kit is designed to detect the AQP1 protein, rather than the AQP1-Abs.

d.ELISA with AQP1 synthetic peptides (extracellular vs. intracellular location of the epitopes)

To differentiate between sera with potentially pathogenic antibodies (i.e., those binding to the extracellular side of AQP1) and those with antibodies to in vivo unavailable epitopes, i.e., to intracellular sites, an ELISA with synthetic peptides corresponding to all extracellular parts of the AQP1 molecule was employed [55]. These included the three extracellular loops (A, C, and E) and three intracellular segments. The ELISA was initially performed with the two separate peptide groups. In agreement with the above-described Western blots with denatured AQP1 that showed AQP1-Abs could also bind to denatured AQP1, all tested AQP1-Ab^pos^ sera were bound to some AQP1 synthetic peptides. Interestingly, most sera were bound selectively only to one peptide group, either the extracellular or cytoplasmic peptides. In competition experiments, in which positive sera were preincubated with a mixture of AQP1 peptides before testing the available free antibodies with RIPA or ELISA using intact AQP1, the binding to intact AQP1 was reduced by more than 50%. This confirmed that most AQP1-Abs bind to the synthetic peptides and therefore the peptide binding findings are representative of the major fraction of the AQP1-Abs repertoire. Using the peptide-ELISA it was found that out of 16 patients with NMOSD phenotype or myelitis and AQP1-Abs, 13 had predominantly antibodies to extracellular loops (mostly to Loop-A) and only three patients had predominantly antibodies to the cytoplasmic peptides. In contrast, all three AQP1-Ab^pos^ patients with classical MS profile (without spinal cord lesions) had only antibodies to the cytoplasmic AQP1 epitopes (i.e., presumably non-pathogenic). These are analyzed in more detail in Section 3.2.

#### 3.1.2. Assays for the Detection of the Autoantibody Binding to the Cell-Embedded AQP1 

Direct CBA

Live-cell CBA (i.e., the test serum is incubated with the transfected cells in their live unfixed state) is currently considered the gold standard method for detecting antibodies to membrane antigens of the nervous system. This approach detects only the antibodies that can bind to the cell-exposed part of the antigen, in its native form, which are obviously more likely to be disease-relevant [60]. The process involves transfecting cells with AQP1-plasmids and control molecules. In the case of Tzartos et al. [55], HEK293 cells were transfected with AQP1-GFP, AQP4-GFP plasmids, and pEGFP-N1, a vector for the fusion of EGFP to the C-terminus of a partner protein. However, probably due to the very low expression of AQP1 molecule in the transfected HEK293 cells (about 100× less, as shown below) Tzartos’ group was not able to develop a reliable live-CBA for AQP1-Abs. Use of secretin in the culture of AQP1-HEK293 transfected cells (and control cells), increased AQP1 expression by about 10×, but it could not be used in the CBA because it dramatically increased background fluorescent staining, which the authors could not eliminate despite their efforts.

Subsequently, the following three different groups also attempted to detect AQP1-Abs in patients with NMOSD phenotype and other CNS demyelinating diseases using different CBA approaches [56,57,58]. Overall, they confirmed the above inability of the CBA with intact cells (live or fixed non-permeabilized) to detect AQP1-Abs.

Long et al. [58] developed an AQP1-fixed-cell CBA with intact or Triton-X-100 permeabilized cells with which they tested sera from patients predominantly with NMOSD phenotype. The fixed-cell CBA without cell permeabilization displayed low efficiency for the detection of AQP1-Abs. However, the use of the permeabilized cells resulted in a dramatic increase in AQP1-Ab detection. With the permeabilized cell-CBA, they tested sera from 249 patients with inflammatory demyelinating lesions, including 98 patients with AQP4-Ab^pos^ and 151 with AQP4-Ab^neg^ sera. AQP1-Ab^pos^ were found in 74.5% AQP4-Ab^pos^ sera (i.e., double positive) and 32.5% of the AQP4-Ab^neg^ sera. Triton-X permeabilization of the cells allows access of the antibodies to the abundant AQP1 molecules inside the cells but also allows access to the cytoplasmic side of the AQP1 molecule. However, as also explained above, antibodies targeting the cytoplasmic side of the molecule are not expected to be pathogenic [61,62] thus reducing the disease-specificity of the permeabilized-cell CBA. 

Gomar et al. [57] developed a fixed-cell AQP1-CBA (with and without Triton-X permeabilization of human AQP1-expressing fixed HEK293 cells). Using this assay, they tested sera from 205 patients from various disease groups. Similar, to the above groups, no AQP1-Ab^pos^ sera were detected with the intact fixed cells. Yet, contrary to Long et al. [58] in cells permeabilized with Triton-X, they observed a noisy signal that made reliable antibody detection impossible.

Finally, Schanda et al. [56] developed a live-cell CBA for AQP1-Abs and applied it to sera from controls and patients with NMOSD or MS diagnosis. AQP1 expression was initially tested by staining with a commercial AQP1-specific antibody, but the antibody did not exhibit a positive signal with live AQP1-expressing cells—only after their fixation with paraformaldehyde. Therefore, to prove the surface expression and the correct topology of AQP1, the authors inserted a myc-tag at the extracellular loop C of AQP1 using site-directed mutagenesis. Staining with an anti-myc-tag monoclonal antibody (mAb) confirmed the surface expression of the mutated AQP1 in live cells. Using CBA with this mutated form of AQP1 (apparently assuming that it did not affect the other AQP1 epitopes), as well as with AQP4, they then analyzed sera from various patient groups for the presence of IgG AQP1 and AQP4-Abs by live-cell CBAs [56] and 81/103 NMOSD patients were found to be AQP4-Ab^pos^. In contrast, similar to the commercial anti-AQP1 mAb, no serum was found positive for AQP1-Abs [56]. Yet, it should be noted that, the candidate sera for the presence of antibodies to cell-surface AQP1 epitopes were only the 22 AQP4-Ab^neg^ NMOSD sera (103-81 AQP4-Ab^pos^). Nevertheless, the authors concluded that there are no AQP1-Abs in NMOSD [56]. Tzartos et al. [63] in a reply to Sandra et al. [56], suggested that the reason of no detection of AQP1-Abs by CBA is probably due to AQP1 being expressed on the surface of transfected HEK293 cells about 100× less than AQP4.

Although all the above four studies acknowledge the difficulty in the development of a live-cell CBA capable of detecting AQP1-Abs in NMOSD sera, overall, they neither confirm nor exclude the presence of antibodies capable of binding to the cell-exposed AQP1.

b.Indirect CBA

Since major difficulties were met during the efforts to establish a reliable and sensitive direct live-cell CBA, an indirect live-cell CBA, which, like the classical live CBA detects only the antibodies capable of binding to the extracellular side of the cell-embedded AQP1, was then developed [55]. 

This assay involves preincubation of the test serum (which may contain AQP1-Abs), with intact live cells transfected with human AQP1 (or AQP4 or control plasmid). This allows only serum AQP1-Abs capable of binding to the extracellular side of the AQP1 molecule in its natural environment, embedded within the plasma membrane of the live cells, to bind to the AQP1-transfected cells. The treated (possibly antibody-depleted) serum, in parallel with the control-treated serum, is then tested by RIPA or ELISA with purified AQP1 to measure the unbound antibodies. The potential reduction of antibodies in the sample pretreated with AQP1-expressing cells (compared to the control treated sample, i.e., that incubated with cells transfected with AQP4 or control plasmid) represents the fraction of the serum’s antibodies capable of binding to the surface-exposed side of the cell-embedded AQP1 (outlined in Figure 4). 

Figure 5 shows the results of an indirect CBA experiment [55]. It demonstrates that HEK293 cells transfected with AQP1-GFP plasmids efficiently removed AQP1-Abs from three of the four test sera (sera no. 1–3), while EGFP- or AQP4-transfected cells had no effect. Serum no. 4 was practically not depleted of its antibodies by the intact AQP1-transfected cells; however, preincubation with detergent-solubilized AQP1-transfected cells efficiently inhibited antibody binding in the RIPA, suggesting that most AQP1-Abs in this serum bind to the cytoplasmic domain of AQP1 [55]. In fact, in AQP1 peptide-ELISA experiments, this serum was bound to the group of AQP1 cytoplasmic peptides whereas sera no. 1–3 bound to the extracellular peptides. AQP1 expression in the cells used in the experiment of Figure 5, was enhanced by the use of secretin in the cell culture, which increased its expression by about 10×. 

It could be argued that among the large numbers of AQP1-expressing cells used in the depletion experiments, a small fraction of them may have been damaged, allowing antibodies directed to intracellular AQP1 epitopes to enter the cells and bind internally. However, the fact that antibodies characterized as cytoplasmic in ELISA with synthetic peptides were not depleted on the same cells but could bind to the AQP1 protein of the solubilized AQP1- expressing cells (serum 4 in Figure 5), practically confirms that AQP1-Abs in sera positive by the indirect AQP1-CBA, indeed bind to the AQP1 molecule in the live intact cells and therefore are potentially pathogenic.

As is described in detail below there was a very good correlation between the indirect CBA and peptide-ELISA results for the tested sera, and a very good correlation with the presence of NMOSD phenotype.

This assay is more laborious than the direct (classical) live-CBA, but it has the advantage over direct CBA, that it is quantitative while it detects exactly the same antibodies with the live CBA. Therefore, it is at least as reliable as the direct CBA. Unfortunately, we are not aware of other groups’ efforts to reproduce this “indirect CBA” approach. 

#### 3.1.3. Combination of Assays for the Detection of AQP1 Antibodies in NMOSD Phenotype

Since a combination of ELISAs with intact AQP1 and with the two groups of synthetic peptides (corresponding to the extracellular and cytoplasmic side of AQP1) corelated well with the results of the more laborious indirect CBA, we routinely screened sera from patients suspected for NMOSD phenotype only with the combined ELISAs. In selected ELISA-positive cases, we confirmed the result with the indirect CBA. Between 2016 and 2022, we conducted screening for the potential detection of AQP1-Abs in all patient samples that were referred to Tzartos NeuroDiagnostics for testing against NMOSD antigens. Our analysis revealed 55/121 (45%) new patients with AQP1 antibodies had antibodies against the extracellular peptide-loops of AQP1, i.e., potentially pathogenic antibodies. All sera with antibodies to the AQP1 synthetic peptides were also bound to the intact AQP1, and all sera bindings to the intact AQP1, were additionally bound to the peptide group. On the other hand, no serum showed binding to both extracellular and cytoplasmic peptides. Only sera positives for the extracellular peptide group were considered positive (AQP1-Ab^pos^). During the same period, we identified, by CBA, 119 new AQP4-Ab^pos^ patients, meaning that the frequency of the AQP1-Ab^pos^ patients was approximately half of that of the AQP4-Ab^pos^ patients. This frequency of AQP1-Ab^pos^ patients is very considerable, which could be invaluable for NMOSD diagnosis. Lastly, no serum was found double positive for extracellular AQP1 and AQP4-Abs. However, we did identify two cases of AQP4-Ab^pos^ sera containing additional antibodies against the cytoplasmic side of AQP1 (unpublished data). 

### 3.2. Are AQP1-Abs in NMOSD Pathogenic?

Studies on the correlation of AQP1-Abs, with CNS astrocytopathy and associated demyelination and on the possible pathogenic role of AQP1-Abs in NMOSD phenotype are limited, and data have been mostly derived by the Athens research team and the collaborating research groups; however, the available results strongly suggest that the AQP1-Abs are disease relevant.

In the initial study [55], both magnetic resonance imaging (MRI) and clinical data were obtained from 22 AQP1-Ab^pos^/AQP4-Ab^neg^ patients. Seventeen of these patients had the clinical core characteristic of myelitis for NMOSD phenotype (LETM *n* = 16, transverse myelitis *n* = 1) with five of them also presenting with optic neuritis. Another five patients had MS phenotype, although two of them had also predominant spinal cord lesions. The clinical and laboratory data of this cohort are summarized in Table 2. The serum samples of these patients were tested for AQP1-Abs using several methods: (a) RIPA (which detected the AQP1-Ab positivity of all these sera), (b) peptide-ELISA, for the differentiation between sera binding to extracellular or cytoplasmic AQP1 segments, and (c) indirect CBA for the detection of antibodies binding to intact AQP1-expressing cells. Interestingly, the majority of the 17 patients with NMOSD phenotype (as well as the two MS patients with predominant spinal cord lesions) had antibodies to the extracellular AQP1 domains and bound to the cell-embedded AQP1, whereas all three patients with classical MS profile had exclusively antibodies to the cytoplasmic side of AQP1 and did not bind on the intact AQP1-transfected cells (Table 2). 

It should be stressed that the 17 AQP1-Ab^pos^ patients with NMOSD phenotype of Table 2 were identified after screening a large number of patients suspected for NMOSD referred by their doctors for NMOSD antibody testing. Given that only a small fraction of NMOSD-suspected patients is expected to have NMOSD, the observation that all patients with antibodies binding to the cell-embedded AQP1 had NMOSD phenotype (or MS profile with predominant spinal cord lesions), strongly suggests that antibodies to the extracellular side of AQP1 could be a potential biomarker for NMOSD among AQP4-Ab^neg^ patients, with a possible pathogenic role. Therefore, we conclude that the AQP1-Ab indirect CBA test is quite competent in “detecting” patients with NMOSD clinical characteristics.

Interestingly, the AQP1-Abs in the three patients with typical MS phenotype (no. 20–22 in Table 2), practically did not bind to the intact AQP1-expressing cells (negative indirect CBA) while they bound to the cytoplasmic AQP1 Loop-B; i.e., these antibodies most probably do not play a pathogenic role, consistent with the absence of NMOSD phenotype-like findings in these patients. The presence of AQP1-Abs in patients with MS phenotype is not surprising, since similar findings have been observed with AQP4-Abs [26,64]. The detection of antibodies to AQP1 cytoplasmic side in patients with non-NMOSD phenotype (like the present three patients with MS-like lesions profile) further suggests that these antibodies are not disease relevant, which is in line with the inability of these antibodies to bind to AQP1 on the intact cells and therefore their potential inability to cause damage to AQP1.

Finally, 3 out of the 22 patients in Table 2 also had a diagnosis of neoplasm suggesting the existence of a paraneoplastic process in some AQP1-Ab^pos^ patients, similar to findings in NMOSD AQP4-Ab^pos^ patients [65]. Although the percentage of these patients with neoplasms in the cohort was considerable (14%), the small number prohibits definite conclusions regarding the frequency of neoplasms in AQP1-Ab^pos^ patients. Nevertheless, it suggests that this percentage may be at least similar to that of the AQP4-seropositive cases.

Two individual case studies of AQP1-Ab^pos^ patients further support the potential pathogenic role of AQP1-Abs [16,49]. Turkoglu et al. [49], in a collaborative study of the groups of E. Tuzun, H. Lassmann, and J. Tzartos, studied a tissue sample from a brain tumefactive demyelinating lesion (TDL) of an AQP1-Ab^pos^/AQP4-Ab^neg^ patient with NMOSD phenotype. Interestingly, in the area of the demyelination (Figure 6a) a dramatic loss of AQP1 was observed (represented by the low immunoreactivity levels) (Figure 6e), while AQP4 expression was preserved (Figure 6d). Also, antibody deposition in astrocytes was observed but without astrocytic destruction and degeneration or neuroaxonal degeneration (Figure 6c), with very limited complement activation in a small area (Figure 6f). Under normal conditions, AQP1 is abundant in subcortical regions of WM and highly expressed on the perivascular foot processes of astrocytes [49], whereas its expression is limited in the cortex of a healthy control GM tissue (Figure 6g). Moreover, AQP1 immunoreactivity was not detected on the astrocytes of the GM (Figure 6h), as in control tissue, while a moderate expression of AQP1 on reactive astrocytes of the WM was reported (Figure 6i) [49]. Additionally, low and diffuse IgG reactivity on perivascular astrocytic foot processes of normal appearing GM (insert of h) was detected and strong IgG reactivity in WM (insert of i) was noted. The pathology seen in this case differed from those reported in the lesions of AQP4-Ab^pos^ NMOSD, where lesions show selective AQP4 loss with preserved AQP1 expression and astrocyte loss [34]. Therefore, the findings of this AQP1-Ab^pos^ case further indicate the possible pathogenic properties of the AQP1-Abs and might link the reduced AQP1 expression expressed by diminished immunoreactive signal with deficient elimination of excess fluids, thus causing the tumefactive lesions, but most likely with a mechanism different from that of AQP4-Abs in AQP4-Ab^pos^ NMOSD cases. In line with this concept was the clinical and radiological improvement that was noted in this patient after plasmapheresis and rituximab treatment, further supporting the disease-relevance of the AQP1-Abs [49]. Τhe pathophysiological relevance of reduced AQP1 presence in WM tissue from patients with NMOSD phenotype remains to be proven.

As described in Section 2.3, AQP1 contains the Colton blood group antigen expressed on erythrocytes [51], while antibodies against Colton antigens can cause hemolytic diseases [52]. Therefore, it has been argued that if AQP1-Abs existed in NMOSD, they might also cause hemolytic disease, which is absent in NMOSD [56]. Yet, this should not necessarily have to happen; for example, although AQP4 is also expressed, in addition to brain and spinal cord, in various organs including skeletal muscles and kidneys, AQP4-Ab^pos^ NMOSD does not usually occur simultaneously with other diseases relevant to these tissues. Nevertheless, as is described below,^,^ another indication of the possible role of AQP1-Abs in both the CNS and RBCs dysfunction is the case of a patient with LETM/MS diagnosis and a history of alemtuzumab treatment, who developed autoimmune hemolytic anemia (AIHA) and a clinical relapse with activity on brain MRI while a high AQP1-Ab titer was measured [16]. Subsequent combined treatment of plasmapheresis and anti-CD20 drug (rituximab) resulted in the full clinical recovery of both diseases with a simultaneous dramatic drop of AQP1-Abs [16]. Furthermore, it was shown that approximately 30% of the patient’s AQP1-Abs could be absorbed in the surface of healthy RBCs, whereas AQP1-Abs were detected in the eluate of the patient’s RBCs as well (Figure 7). 

The high AQP1-Ab titer, in vivo antibody-binding to RBCs during hemolytic anemia and demyelinating relapse, and the drop of AQP1-Ab-titer during the recovery of both clinical syndromes suggest a pathogenic role of these AQP1-Abs. Interestingly, the patient’s antibodies were bound to AQP1 loop-A which contains the Colton group antigens (Coa, Cob), which have been previously linked to hemolytic reactions [51]. AIHA has also been reported following treatment with alemtuzumab (anti-CD52 monoclonal antibody), in at least seven MS cases [66]. All had positive direct Coombs test, i.e., antibodies to RBC surface, but no distinct autoantibody specificity was identified. However, in some AIHA cases, autoantibodies to Colton group antigens (located on AQP1) were detected [51].

Disease transfer models would be highly valuable for demonstration of the pathogenicity of AQP1 antibodies although such evidence is currently not present. There are a few AQP4-antibody mouse models recapitulating major pathological features of NMO [67] enabling scientists to assess the mechanism of AQP4-antibodies in vivo. However, the establishment of a rodent model is a difficult feat to achieve since AQP1 is expressed in rodent brain [10] but not detected on astrocytes [68,69,70], as opposed to AQP1 abundant expression in human astrocytes [32].

## 4. Conclusions

The present review focuses on the reliable detection of AQP1-Abs by a panel of different methods and on the possible relevance of these antibodies with CNS astrocytopathy and associated demyelination. We also discuss the presumed implication of AQP1-Abs in the development of autoimmune disorders, especially in the occurrence of CNS demyelination. Despite controversies in the detection of AQP1-Abs, it seems that the RIPA, the ELISA, and the Western blot methods with purified AQP1 have led to the reliable identification and quantification of these AQP1-Abs, while ELISA with AQP1 synthetic peptides and especially the indirect CBA suggest that some of these antibodies bind to the extracellular side of AQP1.

The detection of AQP1-Abs in some patients with NMOSD phenotype, as well as the binding of AQP1-Abs to the extracellular loops of the AQP1 molecule, are indicative of the possible pathogenicity of these AQP1-Abs. The fact that the majority of the AQP1 antibodies are of the IgG1 subtype [31] further supports their likely pathogenic potential. Data from a cohort with antibodies against the extracellular region of AQP1 in patients with CNS demyelination and NMOSD phenotype showed LETM to be a basic clinical component [55] and in some patients this was also combined with optic neuritis; consequently they had core clinical characteristics of NMOSD.

The two published case reports that demonstrated the pathogenic impact of AQP1-Abs in the CNS and RBCs were also analyzed; specifically, these included (a) the selective AQP1 (but not AQP4) loss in a tumefactive demyelinated lesion in WM of a patient with NMOSD phenotype in conjunction with the antibody deposition in the astrocytes of the WM, and (b) the concurrence of hemolytic anemia and demyelinating relapse with high AQP1 titer, followed by the simultaneous drop of antibody titer and remission of both diseases after “antibody diminishing” therapies (plasmapheresis, rituximab) [16,49].

Future studies are needed to further confirm the role of the AQP1-Abs as biomarkers and pathogens in NMOSD, to characterize in depth the clinical phenotype of the AQP1-NMOSD subtype, and to determine the effect of the known therapies for AQP4-NMOSD cases, versus those for MS, in AQP1-NMOSD cases. Finally, despite the variety of currently developed assays for AQP1-Abs (including two methods for the detection of the potentially pathogenic antibodies to the extracellular side of AQP1), the development of a reliable and easy direct live-cell CBA (possible with the transfection of alternative cell types) would further enhance the use of these antibodies in NMOSD diagnosis and treatment. 

In conclusion, we presented and discussed the presence of antibodies specific for the cell-surface exposed AQP1 domain, in some AQP4-Ab^neg^ patients with NMOSD phenotype, along with their likely pathogenic role. In this context, AQP1-Abs could play a pivotal role, providing a promising diagnostic biomarker for patients with NMOSD phenotype; in the future, these antibodies could also be incorporated into the therapeutic targets of NMOSD, thus leading to novel treatment interventions.

## Figures and Tables

**Figure 1 ijms-24-12982-f001:**
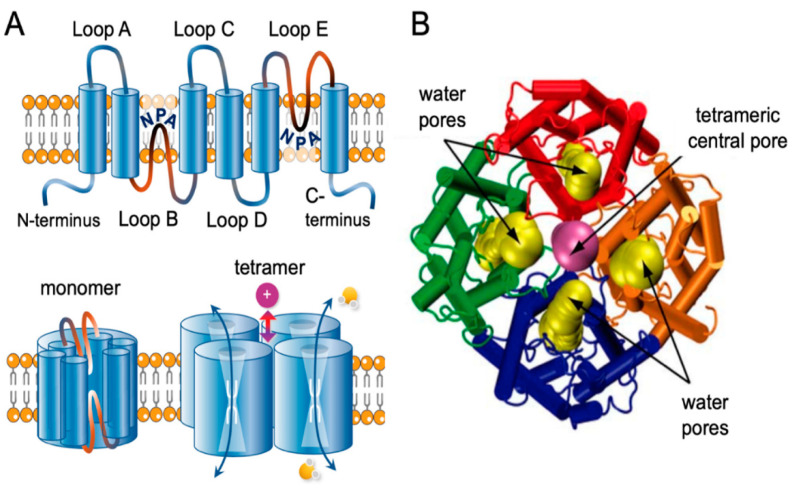
Transmembrane topology of AQP channel and structural view of AQP1. (**A**) Schematic diagram of the structure of an AQP channel depicting the six transmembrane alpha-helices and the five interhelical loop regions A–E. Loops B and E typically fold together within each subunit to form a water pore. Four monomers assemble to form a functional AQP tetramer in the cell membrane. The central pore in a subset of AQPs functions as a gated ion channel. Obtained from [36]. (**B**) Dynamic side view of the structure of the AQP1 channel crossing via the cell membrane. Obtained and modified and reproduced with permission from Elsevier, license #5573061209420 [37].

**Figure 2 ijms-24-12982-f002:**
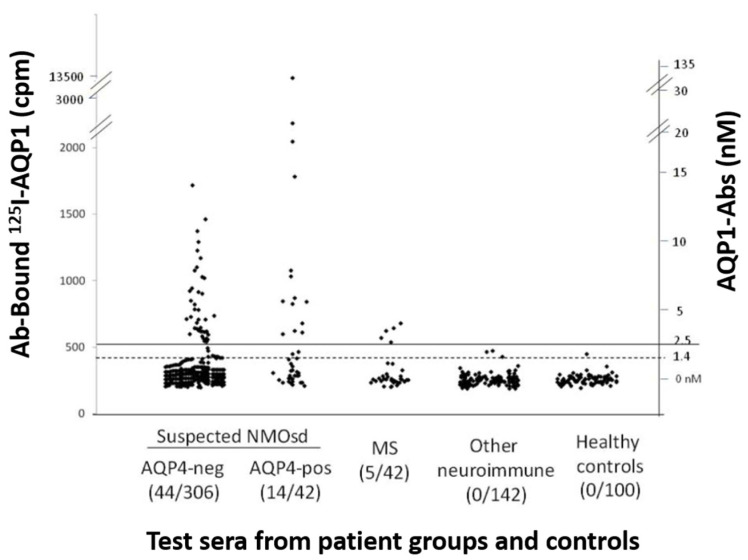
Measurement of AQP1-Abs in the sera of different patient groups and healthy controls by RIPA. The serum samples are (left to right) from patients with suspected ISD phenotype without (AQP4-neg) or with (AQP4-pos) AQP4-Abs, MS, other neuroimmune diseases (131/142 with MG), or healthy controls. The numbers in parenthesis below the group are the number of AQP1-Ab^pos^ sera and the total number of test sera for each group of patients. The dashed and solid horizontal lines denote the cut-off values for ambiguous and positive titers. The left and right y axis show, respectively, the precipitated radioactivity (cpm) and the estimated antibody titer. It must be stressed that RIPAs may detect antibodies to both extracellular and cytoplasmic sides of AQP1, the later apparently being of no pathogenic significance. Modified from [55].

**Figure 3 ijms-24-12982-f003:**
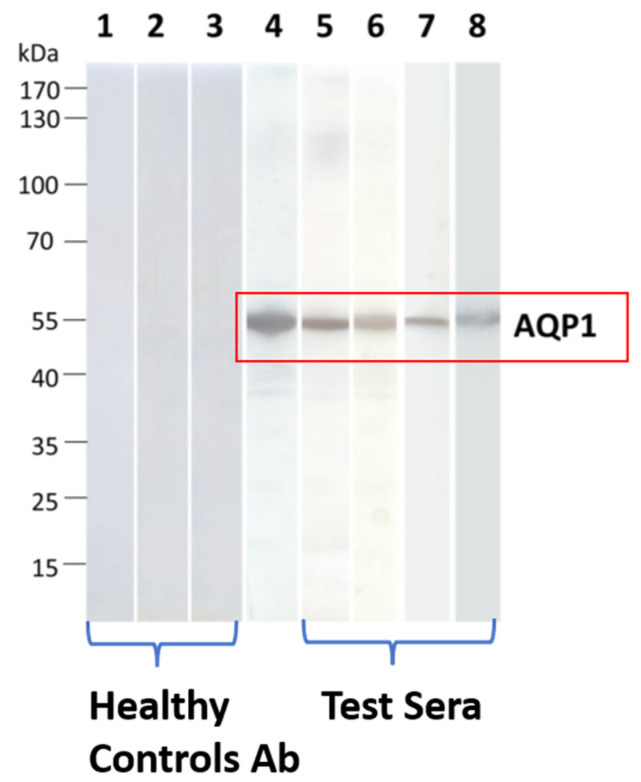
Detection of binding of AQP1-Ab^pos^ sera to SDS-denatured AQP1 by Western blotting. Yeast-expressed AQP1 was electrophoresed and transferred onto nitrocellulose membranes, which were then incubated with the test sera. Lanes 1–3: Three sera from healthy controls); lane 4: commercial rabbit AQP1-Ab; lanes 5–8: four AQP1-Ab^pos^ sera. None of the test AQP1 sera were bound to the control protein MuSK. Modified from [55].

**Figure 4 ijms-24-12982-f004:**
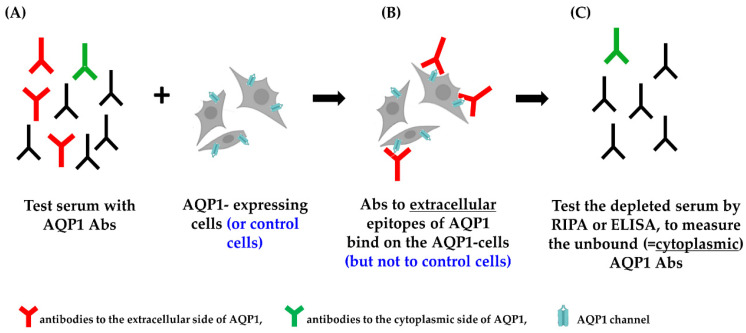
Indirect CBA. (**A**) The test serum is incubated with AQP1-expressing cells. (**B**) Only the antibodies to the extracellular side of the AQP1 molecule bind to the AQP1-expressing cells. (**C**) The treated serum (depleted from the “red” antibodies) is tested by RIPA or ELISA (and compared with the control-treated serum) to measure the unbound antibodies and calculate the percentage of depleted AQP1-Abs, i.e., the % of AQP1-Abs targeting the extracellular side of the AQP1 molecule [55].

**Figure 5 ijms-24-12982-f005:**
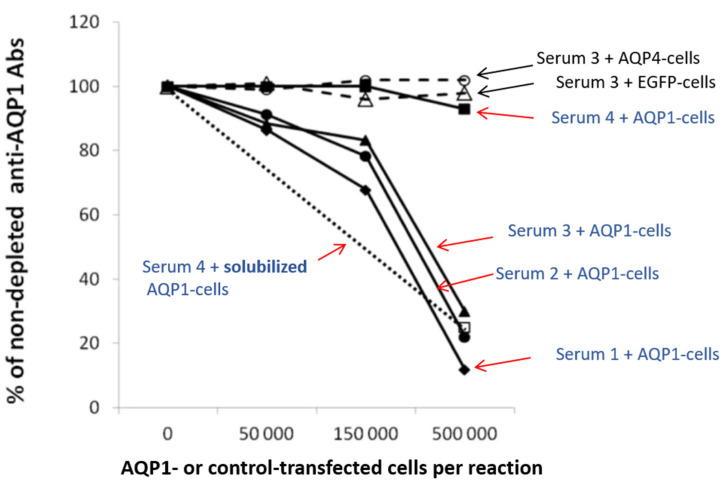
Determination of the percentage of antibodies directed against the extracellular domain of the membrane-embedded AQP1 by “Indirect CBA”. Four AQP1-Ab^pos^ sera were left untreated or were preincubated with increasing numbers of AQP1-GFP (filled symbols) or EGFP- or AQP4-transfected (shown only for serum 3; empty triangle or circle) HEK293 cells treated with secretin to increase surface expression of AQP1 molecule; the untreated and treated samples were tested by RIPA for AQP1-Abs. It is shown that most antibodies from sera 1–3 (which bind to extracellular AQP1 peptides, as determined by ELISA) were depleted by the AQP1-transfected cells but not by the control-transfected cells (with EGFP or AQP4). Serum 4 (which binds to the cytoplasmic AQP1 peptides) as expected, was not depleted, thus confirming the absence of antibodies against the extracellular side of the AQP1 channel; but when it was treated with an extract of AQP1-transfected HEK293 cells (open square), the serum’s antibodies were blocked from binding to the AQP1 in the RIPA. Black arrows show sera treated with control cells without AQP1, whereas red arrows show sera treated with AQP1-cells. Modified from [55].

**Figure 6 ijms-24-12982-f006:**
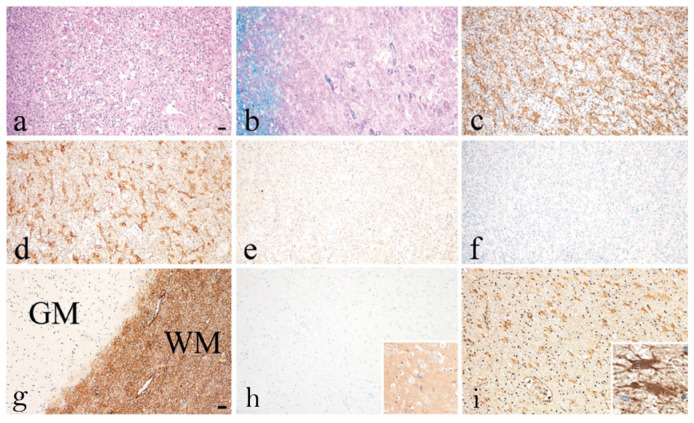
Recurrent tumefactive demyelinating lesions associated with AQP1 immunity. Histopathological images from a biopsy of an AQP1-Ab^pos/^AQP4-Ab^neg^ patient with NMOSD phenotype displaying white matter lesion with (**a**) macrophage infiltration, (**b**) demyelination, (**c**) reactive astrogliosis with GFAP-positive astrocytes expressing (**d**) AQP4 (**e**) but lacking AQP1 immunoreactivity. (**f**) No activated complement was seen after staining for C9neo antigen. (**g**) Normal appearing grey and white matter (WM) section showing the distribution of AQP1 in the brain tissue of a normal control. AQP1 is highly expressed in the WM where it is located on the perivascular foot processes of astrocytes. (**h**) In the present case, no AQP1 immunoreactive signal was detected on the astrocytes of the GM with diffuse IgG reactivity on their foot processes (insert of **h**), (**i**) but there was a moderate expression of AQP1 immunoreactivity levels on perivascular astrocytic foot processes on reactive astrocytes in the WM with IgG deposition (insert of **i**). Magnification bars show 100 μm. Modified from [49].

**Figure 7 ijms-24-12982-f007:**
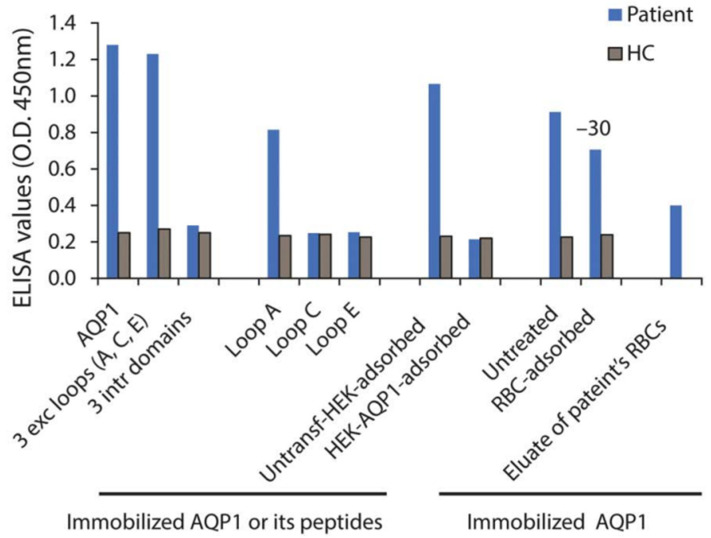
Laboratory results of a patient with concurrent AIHA and demyelinating relapse after alemtuzumab infusion. Serum (from a patient and a healthy control, HC) binding on ELISA wells with immobilized AQP1, peptide mixture corresponding to the AQP1 extracellular and intracellular loops, or the individual AQP1 extracellular loops. “RBC-adsorbed” and “HEK-AQP1-adsorbed” denote serum preincubated with healthy RBCs (group O) or HEK293 cells transfected with human AQP1, respectively, tested for binding on immobilized AQP1. The last bar shows the binding of patient’s RBC eluate to AQP1. Modified from [16].

**Table 1 ijms-24-12982-t001:** AQP function and tissue distribution [1,3,5,6,7,8,9,10].

AQP	Function	Tissue Distribution
QP0	Also known as MIP, major intrinsic protein forms water channels in lens fiber cells and contributes to lens transparency	Lens fiber cells
AQP1	Water transport	Kidney (proximal tubule, thin descending limb of Henle’s loop), Brain (white matter astrocytes, choroid plexus epithelial cells), spinal cord (dorsal horns), optic nerve (corneal endothelium, keratocytes and the ciliary epithelium), inner ear (fibrocytes of the spiral ligament), red blood cells (AQP1 with Colton blood group antigens), lung (alveolar epithelium), and vascular endothelium
AQP2	Water reabsorption	Kidney (principal cells of the collecting ducts)
AQP3	Skin hydration and water transport	Kidney (proximal tubule), skin (basal and supra-basal layers), gastrointestinal tract, and other tissues
AQP4	Water movement across the blood–brain barrier and plays a role in brain edema and water homeostasis	Brain (astrocyte end-feet at blood–brain barrier), optic nerve (retinal glia), spinal cord, skeletal muscle, and other tissues
AQP5	Water transport	Salivary, lacrimal, and other exocrine glands, respiratory tract submucosal glands
AQP6	Possible role in acid–base homeostasis in the kidney.	Kidney (collecting ducts, intercalated cells)
AQP7	Glycerol transport	Adipose tissue, liver, kidney (proximal tubule), and other tissues
AQP8	Transport of water, urea, and other small solutes	Kidney (proximal tubule, thin descending limb of Henle’s loop), liver, and other tissues
AQP9	Transport of water, urea, and other small solutes	Liver (hepatocytes), kidney (proximal tubule), testis, and other tissues
AQP10	Water and solute transport in the gastrointestinal tract and kidney	Tissue distribution: Intestine (colon, ileum, duodenum), kidney (proximal tubule), liver, and other tissues
AQP11	Glycerol channel activity and water channel activity, transport hydrogen peroxide	In several organs such as liver, kidney, and brain
AQP12	Digestive enzyme secretion such as maturation and exocytosis of secretory granules	Pancreatic acinar cells

**Table 2 ijms-24-12982-t002:** Laboratory and clinical data of 22 AQP1-Ab seropositive patients.

Patient ^a^	Sex	Epitopes ^b^	Specific Binding on Live AQP1-Cells (“Indirect CBA”) ^c^	MRI & Clinical Data (& Neoplasms)
1	F	Cytopl	ΝΤ	LETM and optic neuritis (NMO)
2	M	NT	Few antibodies (5%)	LETM and optic neuritis (NMO)
3	F	Extr-C	Most antibodies	LETM and optic neuritis (NMO)
4	F	Extr-A	Most antibodies	LETM and optic neuritis (NMO)
5	F	Extr-A	Most antibodies	LETM and optic neuritis (NMO)
6	F	Cytopl	NT	LETM
7	F	Cytopl	Few antibodies (16%)	LETM
8	F	Extr-E	Some antibodies (30%)	LETM
9	M	Extr-A	Most antibodies	LETM
10	M	Extr-A	Most antibodies	LETM
11	F	Extr-A	Most antibodies	LETM
12	F	Extr-A	Few antibodies (16%)	LETM
13	F	Extr-A	Most antibodies	LETM
14	F	Extr-A	Most antibodies	LETM
15	M	Extr & Cytopl	Few antibodies (19%)	LETM & Hodgkin lymphoma
16	M	Extr-A	Most antibodies	LETM & brain lesions (fulfilled Barkhof criteria)
17	M	Extr-A	Most antibodies	Transverse myelitis & kidney neoplasm
18	F	Extr-E	Most antibodies	MS & predominant spinal cord lesions
19	F	Extr-A	Most antibodies	MS & predominant spinal cord lesions & breast cancer
20	F	Cytopl	No antibodies	MS
21	F	Cytopl	No antibodies	MS
22	F	Cytopl	No antibodies	MS

F, female; LETM, longitudinally extensive transverse myelitis; M, male; MS, multiple sclerosis; NMOSD, neuromyelitis optica spectrum disorder; NT, not tested. ^a^ All 22 patients’ sera tested positive for antibodies to intact AQP1, by RIPA and ELISA, and all tested negative for antibodies to AQP4. In addition, sera were tested by ELISA with AQP1 synthetic peptide (3rd column) and by indirect CBA (4th column). ^b^ Binding of sera to synthetic peptides corresponding to AQP1 segments. Extr-A, -C, -E denote predominant binding to the extracellular AQP1 loops A, C, or E, respectively. Cytopl denotes binding to cytoplasmic segments (loop B, N-terminal or C-terminal end). ^c^ Specific antibody binding to live AQP1-transfected HEK293 cells. Most antibodies denote that 77–100% of the serum’s AQP1-Abs bound to the live cells (i.e., they were depleted from the serum after incubation with the AQP1-expressing cells). Modified from [55].

## Data Availability

Not applicable.

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
