# Peer review of "Potential Role of Antibodies against Aquaporin-1 in Patients with Central Nervous System Demyelination"

_ijms, 2023, doi:10.3390/ijms241612982_

Round 1
Reviewer 1 Report
The authors explained the outline of aquaporin and described in detail multiple measurement methods for AQP1.
Evidence that anti-AQP1 antibodies can cause cell damage in some way is not sufficient to deny the authors’ claim, so there is some likelihood to their argument.
On the other hand, considering the distribution of AQP1, it seems unlikely that AQP1 autoimmunity is a necessity for NMO phenotype.
Further accumulation of disease transfer models and other evidence is necessary to discuss the pathogenicity of anti-AQP1 antibodies.
Reviewer 2 Report
1- NMOSD is recognized as an astrocytopathy and it differs from demyelinating disorders substantially. So, it should be corrected in abstract and manuscript.
2- Why have the authors chosen AQ1 for this study? They should explain why this type of AQs is important.
3- I think the focus of this review should be on the role of AQ1. The authors should discuss AQ4 in the introduction and omit it from section 2.
4- They should discuss the role of AQ1 in the health in separate section. After that, they should explain its role in autoimmune diseases in another section.
5- The most of manuscript has been specified to different methods of AQ1 assays which isn’t related to the main purpose of this study. The authors should review the role of AQ1 in the different diseases and they should just mention the assays in a separate section in brief.
6- The manuscript should be re-written and can not be accepted in this form.
Round 2
Reviewer 2 Report
unfortunately the authors' response has not been attached to the revised manuscript. They should answer the comments word by word. It seems that most of my comments have not been considered in their revision.
Round 3
Reviewer 2 Report
This manuscript is ready to publish.